# Scale Up and Validation of Novel Tri-Bore PVDF Hollow Fiber Membranes for Membrane Distillation Application in Desalination and Industrial Wastewater Recycling

**DOI:** 10.3390/membranes12060573

**Published:** 2022-05-30

**Authors:** Weikun Paul Li, Aung Thet Paing, Chin Ann Chow, Marn Soon Qua, Karikalan Mottaiyan, Kangjia Lu, Adil Dhalla, Tai-Shung Chung, Chakravarthy Gudipati

**Affiliations:** 1Separation Technologies Applied Research and Translation Center (START), Nanyang Technological University—NTUitive Pte Ltd., Nanyang Technological University, Singapore 637141, Singapore; liweikunpaul@ntu.edu.sg (W.P.L.); aungthetpaing@ntu.edu.sg (A.T.P.); chinann.chow@ntu.edu.sg (C.A.C.); ananda.qua@ntu.edu.sg (M.S.Q.); karikalan@ntu.edu.sg (K.M.); adil.dhalla@ntu.edu.sg (A.D.); 2Department of Chemical and Biomolecular Engineering, National University of Singapore, Singapore 117585, Singapore; lukangjia@gmail.com; 3Graduate Institute of Applied Science and Technology, National Taiwan University of Science and Technology, Taipei 10607, Taiwan

**Keywords:** PVDF, membrane distillation (MD), hollow fiber membranes (TBHF), direct contact membrane distillation (DCMD), vacuum membrane distillation (VMD)

## Abstract

Novel tri-bore polyvinylidene difluoride (PVDF) hollow fiber membranes (TBHF) were scaled-up for fabrication on industrial-scale hollow fiber spinning equipment, with the objective of validating the membrane technology for membrane distillation (MD) applications in areas such as desalination, resource recovery, and zero liquid discharge. The membrane chemistry and spinning processes were adapted from a previously reported method and optimized to suit large-scale production processes with the objective of translating the technology from lab scale to pilot scale and eventual commercialization. The membrane process was successfully optimized in small 1.5 kg batches and scaled-up to 20 kg and 50 kg batch sizes with good reproducibility of membrane properties. The membranes were then assembled into 0.5-inch and 2-inch modules of different lengths and evaluated in direct contact membrane distillation (DCMD) mode, as well as vacuum membrane distillation (VMD) mode. The 0.5-inch modules had a permeate flux >10 L m^−2^ h^−1^, whereas the 2-inch module flux dropped significantly to <2 L m^−2^ h^−1^ according to testing with 3.5 wt.% NaCl feed. Several optimization trials were carried out to improve the DCMD and VMD flux to >5 L m^−2^ h^−1^, whereas the salt rejection consistently remained ≥99.9%.

## 1. Introduction

Depletion of fresh water sources combined with the lack of adequate deployment of mature technologies for wastewater treatment and seawater desalination has aggravated the challenges of meeting global water demand for needs such as drinking water, agriculture, and industrial use [1,2,3,4]. Only 3% of the water on Earth is considered fresh water, and only 1.2% has the potential to be used as drinking water because the rest is locked in glaciers, ice caps, and permafrost. To deal with water scarcity and freshwater shortages, seawater desalination processes are widely used. The process of converting seawater to drinking water using thermal or pressure-driven processes is extremely intensive in terms of energy consumption, engineering design, operation, and maintenance. The current solutions utilized for producing drinking water from seawater largely rely on thermal methods, such as multi-stage flash and multi-effect distillation (MSF and MED, respectively) [4,5,6,7], as well as pressure-driven seawater reverse osmosis (SWRO) [7,8], which comprises about 70% of desalination processes worldwide. SWRO has emerged as the most mature and widely accepted technology, with thousands of installations worldwide with a combined capacity of more than 200 million gallons per day, which includes brackish water and seawater systems [9]. However, SWRO systems require extensive energy inputs for high-pressure operation, as well as for operation and maintenance, to mitigate fouling and scaling potential and to improve membrane longevity [8]. In addition, the brine discharged from SWRO systems is very high in salinity and difficult to treat at a higher water recovery rate [10,11]. Another potential freshwater source is wastewater reclamation, where the use of reverse osmosis (RO) alone may not be sufficient for high-strength industrial wastewater. A combination of RO and conventional processes has been proposed, but the process tends to be costly, and the purification may not be as effective [12,13,14]. For these reasons, alternative desalination technologies and the use of renewable energies are being researched and developed to reduce the energy consumption and to improve the overall process productivity [15,16]. Additionally, to improve the sustainability and economic viability of industrial process applications, it is critical that we address not only water reuse but, equally importantly, recovery of key ingredients from the concentrate.

Membrane distillation (MD), with near 100% salt rejection, presents an effective alternative to conventional technologies not just for desalination but also for high-strength industrial wastewater treatment, thereby contributing to maximization of product water recovery, as well as zero liquid discharge. There are many advantages to MD technology in comparison to conventional processes such as RO or MSF, including (1) less harsh operating conditions facilitated by lower requirements for pumping or vacuum pressure; (2) higher rejection of salts, theoretically approaching 100%; (3) fewer chemical dosing requirements for pretreatment; (4) larger contact areas in smaller modular footprints; and (5) ability to treat extremely high salinities of feed water beyond the tolerance limits of seawater RO [17,18,19] (this characteristic is advantageous because, in theory, higher recoveries can be achieved using MD compared to RO); and, finally, (6) use of low-grade heat sources, which opens the possibility of utilizing renewable energy [16,19,20,21,22,23].

MD is a process whereby a highly porous hydrophobic membrane acts as a barrier between the feed stream at higher temperatures and the permeate vapor stream at lower temperatures. The medium used for carrying and/or condensing the water vapor permeate defines the specific mode of MD operations. Direct contact membrane distillation (DCMD) involves a cold liquid flowing on the permeate side that collects and condenses the vapor permeate, using the temperature gradient across the membrane as the driving force [23,24,25]. Vacuum membrane distillation (VMD) involves applying vacuum pressure on the permeate side, along with the hot feed side, using both temperature gradient and pressure gradient as the driving forces, whereby the vapor on the permeate side is collected and condensed externally [26,27]. Other variations include air-gap membrane distillation (AGMD) and sweeping gas membrane distillation (SGMD), whereby the water vapor on the permeate side is collected by air (AGMD) or an inert gas (SGMD), respectively [13,15]. Irrespective of the operating mode, MD is a process based on the change of phase due to a thermal gradient allowing the separation of volatiles. As a result, water from the feed solution follows the principles of vapor–liquid equilibrium, as well as heat and mass transfer to transport across the membrane [21,28].

An ideal MD membrane should be highly porous and hydrophobic, with a very tight pore size distribution and a small pore size, and the membrane fabrication process should be cost-effective and easily scalable from lab to industrial-size equipment. Many advances have been made in developing novel membrane materials in both flat-sheet and hollow-fiber configurations for MD applications [29,30] using polymers such as polyvinylidene fluoride (PVDF), polytetrafluoroethylene (PTFE), and polypropylene (PP) [31,32]. Several other formulations for novel membranes that address pertinent issues such as wetting and fouling in MD applications have been researched at the lab scale, but few have been scaled-up and commercialized for deployment or field validation. A robust invention in membrane technology suitable for MD applications must be scalable on a membrane fabrication line and be conducive to assembly in large industrial-scale modules for field deployment.

In the present study, PVDF was employed as the base material due to its versatility, hydrophobicity, and resistance to a wide range of chemical reagents [22,33] for fabrication of novel tri-bore hollow fiber membranes (TBHF). The rationale behind using PVDF TBHF membranes for MD applications is that the tri-bore structure previously developed by Hua et al. [34], Lu et al. [35], and Luo et al. [36] of Prof. Chung’s group was expected to provide higher mechanical strength, higher liquid entry pressure (LEP_w_) and more surface area per unit volume compared to single-bore hollow fiber membranes [37], leading to higher efficiency. The strategy involves adapting the lab-scale conditions previously developed and optimizing them for a large industrial-scale membrane fabrication unit. The polymer dope composition, coupled with the preparation, is the main obstacle that needs to be overcome for the transition from a bench-scale to a commercial product. A suitable hollow fiber PVDF membrane must have a spongy internal structure with a reduced presence of macrovoids, which are important for avoiding membrane malfunction under high-stress conditions, such as the feed temperature or the vacuum pressure on the permeate side. However, there must be a balance between the membrane pore structure and the specific requirements for the intended applications, as a more open structure (low tortuosity and higher porosity, including macrovoids and pore size) increases flux, whereas the opposite is important to prevent membranes wetting and failure [38,39].

In the present study, we report the translation of novel PVDF tri-bore hollow fiber membranes from lab-scale to industrial-scale batches with dope sizes of 1.5 kg, 20 kg, and 50 kg. To demonstrate the robustness and the feasibility of production of membrane modules on an industrial scale, small 0.5-inch diameter modules with 6–10 fibers, as well as 2-inch diameter modules with >200 fibers, were fabricated, and the MD applications were validated with testing conditions in DCMD and VMD modes. Module evaluation was performed using a feed solution of 35 g L^−1^ NaCl to simulate seawater desalination conditions. 

## 2. Materials and Methods

### 2.1. Materials

The following chemicals used during for membrane fabrication and scale up were of industrial and reagent grade and used without further purification: polyvinylidene fluoride (PVDF, Kynar HSV 900 PWD resin, Arkema, Calvert City, KY, USA), lithium chloride (LiCl) (GCE Laboratory Chemicals–TACT Chemie S.E.A. Pte. Ltd., Singapore). dimethylacetamide (DMAc) (Puyang Guangming Chemicals Co. Ltd., Puyang City, China); ethylene glycol (EG) (TACT Chemie S.E.A. Pte. Ltd., Singapore); methanol (MegaChem Ltd., Singapore), HPLC-grade hexane (Fisher Scientific, Fairlawn, NJ, USA), sodium chloride (NaCl) (Pure Dried Vacuum Salt, INEOS Enterprises, Runcorn, UK). Deionized water was acquired from a PURELAB Option-Q DV 25 unit from ELGA with a resistivity of 18.2 MΩ·cm.

### 2.2. Fabrication of PVDF Tri-Bore Hollow Fiber Membranes

PVDF hollow fiber membranes were fabricated with the polymer dope formulation and spinning conditions developed by Lu et al. [35]. The hollow fibers were fabricated using a dry–wet jet phase inversion spinning process using a custom-designed tri-bore spinneret with a circular geometry, the specifications of which are detailed in [36]. Table 1 summarizes the dope formulation and spinning parameters, such as line speed, air-gap distance, dope flow rate, and bore fluid flow rate, and includes temperatures of the dope, bore liquid solution, and the coagulation bath. However, some of the conditions for the dope mix and membrane spinning were modified to adapt the process to large-scale production with consistent results.

Briefly, the solvent was added into a dope mixing tank equipped with thermometers, inlet valves for (a) solid and liquid addition and (b) nitrogen, outlet valves for venting and vacuum suction, and a monitoring window. Initially, a portion of the solvent (DMAc) was introduced into the tank, followed by sequential addition of LiCl and EG for solubilization. Additional DMAc was added to the mixture and stirred at a given mixing propeller speed (~25 Hz) until the solution was clear and no particulates were visible. PVDF was then added slowly in batches to avoid formation of lumps in the tank. After all the PVDF was added to the solution, the remaining DMAc was added, and the tank was sealed with an air-tight lid. The mixing speed was increased to 30 to 40 Hz, and the temperature was initially set at 35–40 °C. The solution temperature was constantly monitored using a probe inside the tank. As the solution temperature increased, due to the exothermic heat of mixing, the solution was stirred until the temperature was stabilized. Subsequently, the set temperature was increased gradually in increments of 5 °C until the solution temperature reached between 80 and 85 °C. After a clear solution could be seen from the monitoring glass window, stirring was stopped, and the temperature was stabilized. The solution was degassed under a static vacuum pressure of ~−1 bar for 3 days until spinning. During spinning, the fibers were collected by winding around a circular winding wheel of 2 m circumference at an initial speed of 7.8 rpm. As the number of fiber layers winding around the wheel reached 3 or 4, and an increase in the line tension was visibly observed, the winding wheel rotation speed was reduced to 7.2 or 7.3 rpm to minimize the line tension, and the fibers were cut. After cutting the fibers from the circular winding wheel, they were rinsed in water for 24 h to remove residual solvents. The membranes were then soaked in methanol for 1 h to remove excess water. The procedure was repeated twice more with fresh methanol. The methanol-rinsed fibers were then soaked thrice in fresh hexane for a period of one hour each time. After hexane soaking, the membranes were dried in a dry room with controlled humidity and temperature for at least 24 h before inspection and selection for module fabrication.

The temperature of the polymer dope was constantly monitored during mixing, degassing, and spinning. The spinning required up to three working days for dope batch sizes ≥20 kg, which required degassing at the end of each working day. The viscosity of all dopes was measured close to the spinning temperature of 80 °C using a viscometer (Cole-Palmer VCPL 340015, Vernon Hills, IL, USA).

### 2.3. Membrane Characterization

The dried membranes were visually inspected under an optical microscope (Leica DVM6 optical microscope, Wetzlar, Germany), and the images were used to measure the outer diameter (OD) and the inner diameter (ID) using image processing software. The membrane morphologies were characterized with a field emission scanning electron microscope (FESEM) (JEOL JSM-7200F) operated at a 5.0 kV accelerating voltage. The non-conducting PVDF surfaces were sputter-coated with platinum using a JEOL JFC-1100E ion-sputtering device before measurement.

The pore size distribution was determined by a capillary flow porometer (CFP 1500AEX, Porous Material. Inc., Ithaca, NY, USA), the working principle of which was based on bubble-point and gas permeation tests. The hollow fiber samples were potted into sample holders and soaked with a wetting fluid (Galwick) with a surface tension of 15.9 × 10^–3^ N m^−1^ until completely wet. During the test, the gas flow rate was increased stepwise and passed through the saturated sample until the applied pressure exceeded the capillary attraction of the fluid in the pores. By comparing the gas flow rates of both wet and dry samples at the same pressures, the percentage of flow passing through the pores larger than or equal to the specified size can be calculated from the pressure–size relationship. The mechanical properties of hollow fiber membranes were examined using a universal tensile tester (Instron 3342, Norwood, MA, USA). Each specimen was firmly clamped by the testing holder and pulled longitudinally at an elongation rate of 50 mm min^−1^ at room temperature. The corresponding mechanical properties were determined by the built-in software.

The contact angle was determined using a tensiometer (DCAT11 Dataphysics, Filderstadt, Germany). The contact angle quantifies the wettability of a solid surface by a liquid. The sample was inserted into an electrobalance for cyclical immersion in DI water. The contact angle was calculated from the wetting force using the Wihelmy method. The overall porosity of membranes was determined by the gravimetric method with Equation (1):(1)$$Porosity=1-\frac{Volume_{polymer}}{Volume_{total}}=\left(1-\frac{Membrane\;weight\;\left(g\right)\;/\;Membrane\;volume\;\left(cm^{3}\right)}{Polymer\;density\;\left(g\;/\;cm^{3}\right)}\right)\;\times \;100\%$$
where the PVDF density was 1.78 g cm^−3^ and the membrane volume was calculated based on the OD and the ID of the fibers.

LEP_w_ was determined using dead-end hollow fiber modules containing a single membrane fiber. LEP_w_ measures the pressure required to force water through the pores of a dried membrane and is an indication of how easily a hydrophobic membrane can be wetted. Water was gradually pressurized at an increment of 0.5 bar. As water was pressurized, it could be pushed across the membrane pores, and the pressure at which water droplets were visible on the outer surface of hollow fibers was recorded as the LEP_w_ of the membranes.

### 2.4. Membrane Module Testing

The tri-bore hollow fiber membranes fabricated in 20 kg and 50 kg batch sizes were assembled into membrane distillation (MD) modules of two different sizes, i.e., 0.5-inch diameter and 2-inch diameter, as shown in Figure 1. The modules were evaluated for MD performance in both direct contact membrane distillation (DCMD) and vacuum membrane distillation (VMD) modes.

DCMD tests were conducted using the laboratory setup depicted in Figure 2a. A saline solution comprising 3.5 wt.% NaCl was prepared as the feed and heated to the requisite temperature using a heating circulator. In the out-to-in testing mode, the feed was circulated through the shell side of the membrane with a centrifugal pump at the requisite flow rate and temperature. Simultaneously, a permeate solution (DI water) at 10–11 °C, by means of a refrigerated circulator (RT7, Thermo Scientific, Waltham, MA, USA), was circulated along the lumen side of the membrane using a rotary pump at the requisite flow rate. The feed and the permeate streams were configured for concurrent directions. The temperatures of both feed and permeate streams were also constantly monitored.

The system was allowed to run for 30 min until the set temperatures were reached; then, the weight change of the permeate solution was recorded. The conductivity of the permeate solution was measured by an electrical conductivity meter (Lab 960, Schott, Mainz, Germany) dipped into a beaker that contained the permeate solution. In order to accurately detect salt leakage, the volume of the permeate solution inside the beaker was controlled below 300 mL by transferring the excess water back to the feed tank. For performance comparison, DCMD tests were also conducted by circulating the feed along the lumen side under the in-to-out testing mode. The testing conditions for the tri-bore hollow fiber membrane module are detailed in Table 2.

### 2.5. Vacuum Membrane Distillation (VMD) Mode

The hollow fibers were assembled into 0.5-inch or 2-inch diameter modules, as shown in Figure 1, and tested at the Environment & Water Technology Centre of Innovation (EWTCOI), Singapore. Under the in-to-out VMD mode, the feed water was recirculated through the lumen side of the hollow fibers. The liquid feed entered the module in an upward direction to minimize air bubbles in the module. Once the feed inlet temperature in the membrane module reached a steady state, the vacuum pump was switched on to apply vacuum pressure on the shell side of the hollow fibers. The timer for permeate collection was started, and the permeate was collected by condensing the water vapor either in an ice-chip bath, which was periodically refilled with ice chips (for 0.5-inch modules), or using a chiller (for 2-inch modules) at 15 °C. The amount of permeate collected was gravimetrically determined using a weighing scale, and electrical conductivity (EC) was also measured. The flow paths of the feed and the permeate were reversed for the in-to-out and the out-to-in testing modes, as shown in Figure 2b.

The water flux in L m^−2^ h^−1^ was calculated as shown in Equation (2):(2)$$Flux=\frac{Permeate\;volume\;\left(Litres\right)}{Membrane\;area\;\left(m^{2}\right)\;\times \;Duration\;\left(h\right)}$$

Salt rejection was determined using Equation (3):(3)$$Salt\;rejection=\left(1-\frac{C_{p}}{C_{f}}\right)\;\times \;100\%$$
where *C_p_* is the concentration of the permeate solution, and *C_f_* is the concentration of the feed solution. 

## 3. Results and Discussion

### 3.1. Validation of Membrane Fabrication Process on Pilot-Scale Equipment

The objective of this study was to scale up the fabrication process for tri-bore hollow fiber membranes from small lab-scale batches of ~1 kg to 50 kg dope size. Before the dope quantity could be scaled up from 1.5 kg to industrial-scale batches, the lab process was replicated on a pilot spinning line for reproducibility and validation. The dope formulation was optimized on a lab-scale hollow fiber spinning line with spinning conditions reported in [35]. In this study, the optimized process was adopted to suit our pilot-scale membrane fabrication setup. Initially, the tri-bore hollow fiber spinning process was validated on small-batch sizes of 1.5 kg to reproduce the membrane properties previously reported in [35]. The pilot-scale fabrication line is different from the lab-scale equipment in terms of pump type, as well as the size of the dope tanks, coagulation bath, and rinsing bath, all of which might influence the quality of the produced membranes. In order to maintain the process conditions, such as dope flow rate and bore liquid flow rate, similar to those previously reported [35,36], the pumps of the pilot-scale fabrication line were calibrated prior to use (Appendix A).

#### Membrane Characterization

More than 10 batches of 1.5 kg size were prepared, and membranes were fabricated using the spinning conditions detailed in Table 1. Membrane quality and reproducibility were evaluated by randomly choosing fiber samples selected from each batch. At least six fibers were selected from each batch; their inner diameter (ID) and outer diameter (OD) were determined using an optical microscope fitted with a digital camera, and the images were analyzed using the accompanying software. A typical image used for determination of OD and ID is shown in Appendix A. Figure 3 displays a variability plot of the OD and ID (of three bores) over five batches of 1.5 kg size. The OD for each batch, which corresponds to each data point, was found to be consistently within the range. Only 6% and 3% of data points were outside the upper specification limit of 2.0 mm and the lower specification limit of 1.5 mm, respectively. Similarly, the ID values of the three bores fell within two ranges: (1) two bores in the range of 0.25 to 0.4 mm and (2) one bore in the range of 0.5–0.6 mm.

The spinneret dimensions are shown in Appendix A, with the diameter of the three bores being 0.5 mm. The deviation from the set diameter of 0.5 mm presumably stems from minor variations in the bore liquid flow rate and dope flow rate, as well as small differences in their phase inversion processes during the spinning operations. 

The porosity, contact angle, liquid entry pressure, and DCMD flux of the TBHF membranes prepared in 1.5 kg batch sizes are summarized in Figure 4. To evaluate the hydrophobicity of the TBHF membranes, the water contact angle values were evaluated on at least five or six fibers from each batch, and least three measurements were performed on each fiber sample for statistical significance. It is evident from Figure 4 that a water contact angle of 80°–90° was observed for all fibers, indicating the hydrophobic nature of the PVDF membrane surface. 

The bulk porosity of the TBHF membranes consistently remained within the range of 70–80%. These results are in agreement with the values reported in the literature for similar dope compositions [35,40,41]. The liquid entry pressure (LEP_w_) is a critical parameter for membrane distillation applications that determines the antiwetting potential of the membranes at a given operating pressure. The LEP_w_ of the TBHF membranes was consistently within the range of 2.5–3 bar for all samples tested across five batches (Appendix A). Membrane performance was evaluated in DCMD mode using a 3.5 wt.% NaCl solution as the feed. As part of the technology scale-up process and feasibility assessment for reproduction of the membrane quality on a large pilot-scale membrane fabrication unit, the membranes were assembled into small lab-scale modules of 0.5-inch diameter for DCMD tests in out-to-in mode unless otherwise specified. The module specifications and the test conditions are detailed in Table 2. The DCMD flux for the modules was prepared from five 1.5 kg batches is in the range of 13 to 14 L m^−2^ h^−1^, indicating repeatability of the membrane quality across several batches, as also demonstrated by the contact angle, porosity, and LEP_w_ data (Figure 4 and Appendix A). The population mean values (i.e., the average of all data points from all samples) and the small standard deviations for contact angles (82.1° ± 6°), porosity (72.8% ± 2.5%), LEP_w_ (2.6 ± 0.1 bar), and DCMD flux (13.6 ± 0.4 L m^−2^ h^−1^ validated the successful process transition from lab-scale to pilot-scale equipment.

### 3.2. Translation of Membrane Fabrication Process from a Lab Scale to an Industrial Scale

The successful validation of the membrane fabrication parameters using a small 1.5 kg dope quantity led us to scale up the batch size to large-scale fabrication using 20 kg and 50 kg batch sizes. Whereas the primary difference in large-scale fabrication was the use of larger dope mixing tanks, other equipment changes, such as larger pumps and tubing, were implemented as required. Other process variables, such as the dope temperature, mixing time, degassing time, and flow rates for dope and bore liquid, were maintained at the values shown in Table 1.

#### 3.2.1. Characterization of TBHF Membranes Prepared on an Industrial Scale

The surface morphologies of the TBHF membranes fabricated in large-scale batches of 20 kg and 50 kg were evaluated using field emission scanning electron microscopy (FESEM). FESEM images of the cross section, inner surface, and outer surface are shown in Figure 5. The SEM images of the hollow fiber cross section (Figure 5a,e) show uniform bores in membranes made from 20 kg and 50 kg batch sizes. The cross-section images of a fiber wall at higher magnification (Figure 5b,c) show a very similar structure with small finger-like macrovoids at the end of the wall and a mostly sponge-like dense, porous layer across the rest of the membrane. The inner surfaces were found to possess uniform porous morphologies, and the outer surfaces were denser. The morphologies observed for the large batches were similar to those of lab-scale batches reported in [35].

The porosity data of the PVDF TBHF membranes prepared from different batch sizes are shown in Figure 6a. Whereas membrane porosity showed a reproducible trend within a 70–75% range for all batches spun from 1.5 kg and 20 kg sizes, more variability was observed for the 50 kg batches. The membranes spun from the two initial batches of 50 kg size possessed lower porosity of 60–65%, whereas those from the subsequent batches had a higher porosity. This variation may arise from the effect of spin-line stresses on chain packing during spinning. For the first two spinning trials, the fibers were allowed to continuously wrap around an industrial-scale winding drum with a circumference of 2 m until several layers of hollow fibers accumulated. This may result in considerable compression and elongation stresses; thus, the resultant fibers had a tighter morphology and a lower porosity. In the subsequent runs, the fibers were cut from the winding drum after reaching a maximum of three layers of fibers in order to preserve the fiber structure. As a result, a porosity of 75–80% was consistently achieved in the remaining spinning trials. The tensile stress of the 1.5 kg batch membranes was lower, with a much wider variability than the 20 kg or the 50 kg batch membranes. The tensile stress values (Figure 6b) observed for the 20 kg and 50 kg batches are more consistent with previously reported values.

Table 3 summarizes and compares other properties, such as the dimensions, contact angle, liquid entry pressure, pore diameter, DCMD flux, and salt rejection for membranes prepared from different batch sizes and assembled into 0.5-inch modules. As expected, the outer diameters (ODs) of the TBHF membranes were within the range of 1.7 mm to 1.9 mm. The inner diameter (ID) is reported as the average of three IDs of the three bores. Their standard deviations were 17%, 15%, and 6% for the 1.5 kg, 20 kg, and 50 kg batches, respectively. The higher consistency over the five 50 kg batches indicated that the hollow fiber fabrication process was successfully scaled-up. The contact angle values were in the range of 80–90°, indicating the hydrophobic nature of the PVDF materials, as required for MD applications. The liquid entry pressure (LEP_w_) is a key parameter in determining the pressure at which the feed water can enter the membrane pores, along with the vapor. Membrane pore wetting is a common challenge in MD applications that results in loss of salt rejection and diminishes the permeate quality [32]. LEP values of 2.8–4.0 bar for all the PVDF TBHF membranes prepared from different batch sizes are significantly higher than the operating pressure difference between the feed and the permeate <1 bar, indicating the robustness of these membranes during MD operation. The DCMD flux and salt rejection for all the membranes were evaluated using a custom-built MD unit, as depicted in Figure 2, and tested under the out-to-in mode, as detailed in Table 2. The permeate flux remained >10 L m^−2^ h^−1^ for all the membranes, and the salt rejection consistently was ≥99.9% over a testing period of 1–2 h, without pore wetting or compromising the permeate quality, indicating the high efficiency of the membranes.

As part of our objectives to scale up the membranes from lab-scale to industrial-scale modules for pilot-scale validation of applications, modules with 2-inch diameter were also fabricated using membranes spun from different batch sizes. The preliminary data for the 0.5-inch and 2-inch modules prepared from 50 kg batches are presented in Table 4.

As shown in Table 4, as the module size increases from 0.5-inch to 2-inch, the MD fluxes in both the DCMD and VMD modes drop significantly from ~≥10 L m^−2^h^−1^ to ~1 L m^−2^h^−1^. The 0.5-inch modules contain 6–10 fibers and have an effective length of 100–120 mm for DCMD and VMD tests, as shown in Table 2. Although the salt rejections for both testing modes remain consistently over 99.99%, the flux for VMD mode is higher (12.4 L m^−2^h^−1^) than for DCMD mode (10.6 L m^−2^h^−1^). VMD mode has been known to produce a higher flux than the other modes of MD due to lower conductive heat loss along the module length, as reported multiple times in the literature [42,43]. More importantly, the MD flux is reduced by more than 80% for the large 2-inch modules, presumably due to several factors. Longer modules have been reported to result in significant flux loss due to extended temperature polarization [44] along the feed path for both DCMD and VMD modes, whereas DCMD mode is subject to additional factors, such as conductive heat loss along the membrane cross section [45]. In addition, a significantly larger number of hollow fiber membranes is packed in 2-inch diameter modules than in the 0.5-inch modules (>200 vs. ≤10 in 0.5-inch), which may lead to suboptimal flow distribution on the cold-permeate side. As a result, the enlarged membrane area in 2-inch modules is inadequately utilized, leading to a diminished vapor flux through the membrane under DCMD mode. Dudchenko et al. reported that the permeate flux dropped by more than 40% when the module length was increased from 4 cm to 20 cm, independent of membrane characteristics. In the same study, it was also observed that the permeate flux decreased with increasing membrane area [46].

#### 3.2.2. MD Flux Optimization

In order to improve the MD performance of the PVDF TBHF membranes, several module variables were manipulated, and the permeate flux was measured in both DCMD and VMD modes using the conditions presented in Table 2. Several factors were studied, such as module length, number of fibers in a module, effective membrane area, feed temperature, permeate-side vacuum pressure, feed and permeate flow rates, etc. To this end, several 2-inch modules were prepared with characteristics detailed in Table 5. Because most of the membrane properties were standardized in different batch sizes, the membranes used for the flux optimization studies were randomly selected from different 50 kg batches and assembled into 2-inch modules. 

##### DCMD Flux Optimization

The effect of several module parameters was examined, such as the number of fibers in a module, effective length, and membrane area, as well as experimental conditions, such as feed flow rate and flow configuration (out-to-in vs. in-to-out). 

As shown in Figure 7, the longest modules (440 mm length, SN-1 and SN-2) had the lowest observed flux for all tested conditions. The flux increases by more than 50%, from ~1.5 L m^−2^ h^−1^ to ~2 L m^−2^ h^−1^ (SN-3), when the module length was decreased from 440 mm to 240 mm. Interestingly, the flux increased by ~95%, from ~2.4 L m^−2^ h^−1^ to ~4.8 L m^−2^ h^−1^, when the module length was further decreased to 200 mm (SN-5). Clearly, there was a considerable difference in flux value when the module length is increased from 200 mm to 230 mm. The maximum flux resulting from the optimization trials was 4.8 L m^−2^ h^−1^. More studies are required in the future to identify the key factors affecting the DCMD flux as a function of the module length. As explained before, for a given membrane area and module diameter, the module length contributes significantly to the conductive heat loss across the length that results in a flux decline [44,46].

#### 3.2.3. VMD Flux Optimization

Several testing conditions were evaluated under VMD mode to optimize the flux for the 2-inch modules. Three different flow configurations, i.e., in-to-out, in-to-out (double outlet), and out-to-in, were evaluated in conjunction with several operating conditions, such as vacuum pressure, feed flow rate, and effective membrane area. The in-to-out (double outlet) condition involves opening both the outlet ports during operation in order to increase the permeate flow in comparison to a single-outlet, which is usually used (Figure 2b). The VMD results are categorized into groups 1, 2, and 3 according to the testing modes of in-to-out, in-to-out (DO), and out-to-in, respectively. Within group 1, it is apparent that the flux declined as the module length increased from 240 mm to 440 mm, which is in line with the results reported in the previous sections. In group 3, the flux was generally lower than that of in-to-out mode. However, module length and membrane area were still the main factors affecting the performance. Interestingly, for a given module length (either 240 mm or 440 mm), decreasing the pressure (or increasing the vacuum) from −0.3 bar to −0.7 (or −0.75) bar did not improve the flux as expected from increased driving force for vapor permeation at a higher vapor pressure. Among these three groups, group 2 had the highest flux of 5–7 L m^−2^ h^−1^ when the feed flow rate was 6 L min^−1^ and the vacuum pressure was −0.8 bar. Although the highest flux of ~7 L m^−2^ h^−1^ achieved in the VMD optimization tests for 2-inch modules was lower than that (12.4 L m^−2^ h^−1^) observed for the 0.5-inch modules, salt rejection remained consistently over 99.99% during the testing period of 1.5–2 h, indicating the efficiency and wetting resistance of the membranes. Nevertheless, the key observations regarding the correlation among module parameters, operating conditions, and MD flux of the PVDF TBHF membranes, as summarized in Figure 7 and Figure 8, form the basis for further research on optimization of the MD performance of 2-inch, 4-inch, and 8-inch module sizes. The systematic statistical design of experiments (DOE) to evaluate the influence of several other factors, such as packing density, flow configuration, feed, and permeate flow rate on the MD flux shall be the subject of future research and publications.

## 4. Conclusions

Novel tri-bore PVDF hollow fiber membranes were scaled-up from a lab-scale process to an industrial-scale fabrication process with consistent membrane characteristics reproduced from 1.5 kg, 20 kg, and 50 kg batches. The membranes were then assembled into small 0.5-inch modules for evaluation under DCMD and VMD modes for desalination applications. Very high fluxes ≥10 L m^−2^ h^−1^ were observed for 0.5-inch diameter modules, indicating a successful scale up of the membrane fabrication process. Larger pilot-scale modules with 2-inch diameter and membrane areas ranging from 0.3 to 0.7 m^2^ were fabricated and tested under DCMD and VMD modes. The 2-inch module had a significantly lower flux (~1 L m^−2^ h^−1^) than the 0.5-inch modules (≥10 L m^−2^ h^−1^). Several module parameters, such as packing density and module length, as well as operating conditions, were evaluated, resulting in an optimal flux of 7 L m^−2^ h^−1^ in VMD mode. Salt rejection remained ≥99.9% throughout the testing duration, indicating the applicability of the membranes in seawater desalination, as well as high-strength industrial wastewater treatment for water reuse and resource recovery, in addition to zero liquid discharge applications. The salt rejection ≥99.9% consistently observed for the different module sizes and in both the DCMD and VMD testing modes is consistent with the typical MD purification efficiencies reported in the literature, as well as in comparison to commercially available polytetrafluoroethylene (PTFE) and polypropylene (PP) hollow fiber membranes. The optimal flux of ~8 L m^−2^ h^−1^ is slightly lower than the flux values of 10–13 L m^−2^ h^−1^ reported for commercial membranes. There is significant room for improvement of flux through optimization of the process variables, such as the temperature differential (DCMD) and vapor pressure differential (VMD), along with other factors, such as the feed flow rate, module packing density, and the fiber packing configuration. These observations shall form the basis for a more thorough statistically based optimization of MD performance starting with 2-inch and, subsequently, for 4-inch and 8-inch module sizes. Future studies shall focus on (a) evaluation of the fouling potential, scaling potential, and flux recovery ratio profile; (b) long-term evaluation of the membrane performance in DCMD and VMD modes on a large-scale pilot unit with a capacity of 5000 L/day to evaluate the wetting propensity, fouling/scaling potential, cleaning requirements, cleaning frequency, and the retention of desalination performance over a period of 3 to 6 months; and (c) benchmarking of the tri-bore hollow fiber membrane module performance against commercial hollow fiber MD membranes for techno-commercial analysis as a path towards commercialization. 

## Figures and Tables

**Figure 1 membranes-12-00573-f001:**
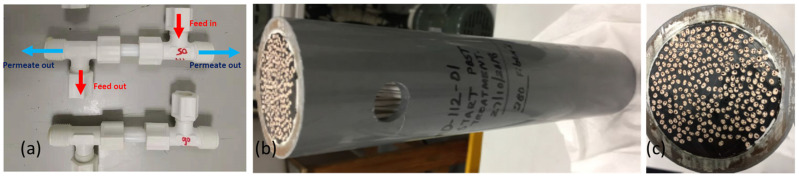
Scale up of tri-bore hollow fiber membrane modules. (**a**) Lab-scale testing modules (0.5-inch diameter), (**b**,**c**) pilot-scale modules (2-inch diameter).

**Figure 2 membranes-12-00573-f002:**
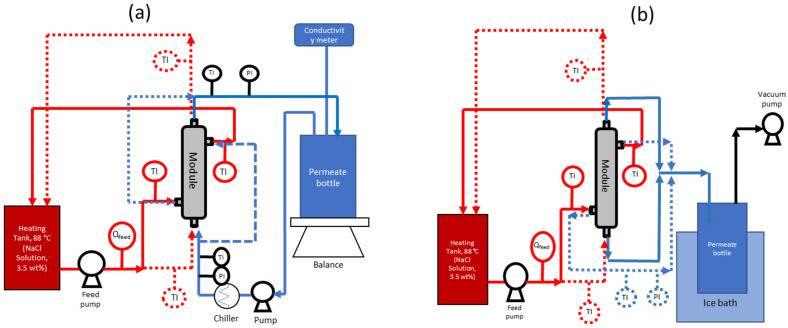
Schematic of the membrane distillation test skids operated in (**a**) direct contact membrane distillation (DCMD) mode and (**b**) vacuum membrane distillation (VMD) mode. The red lines indicate the flow path of the hot feed (salty water), and the blue lines indicate the flow path of the cold permeate. The solid lines in both figures indicate operation under the “out-to-in” configuration, and the dotted lines indicate operation in “in-to-out” configuration.

**Figure 3 membranes-12-00573-f003:**
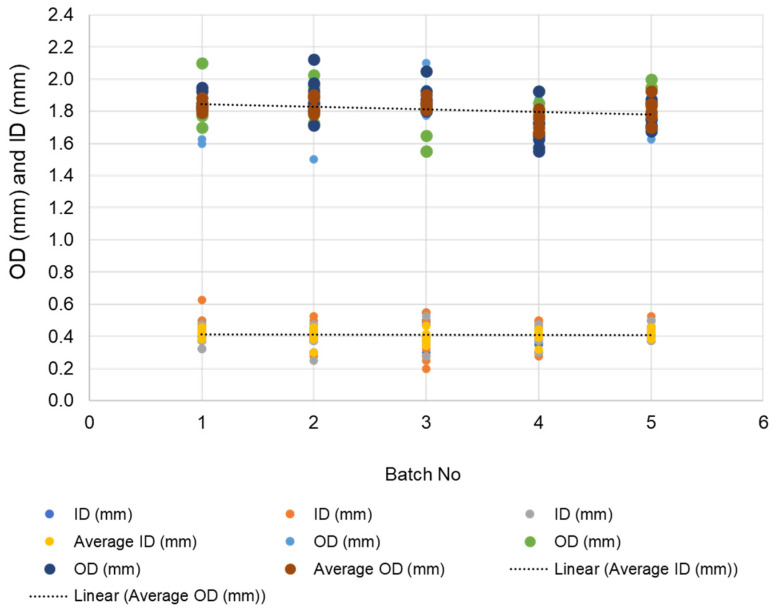
Outer diameter (OD) and inner diameter (ID) of the three bores, along with the average of OD and ID of the tri-bore hollow fiber membranes.

**Figure 4 membranes-12-00573-f004:**
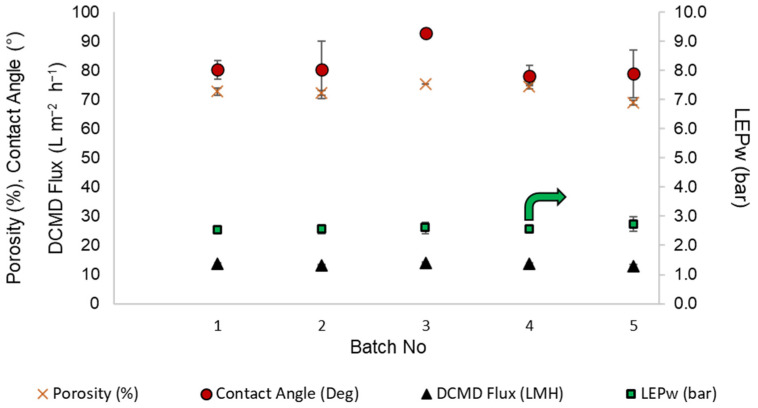
Porosity (%), contact angle (°), DCMD flux (L m^−2^ h^−1^), and liquid entry pressure (LEP_w_, bar) of different batches of 1.5 kg size.

**Figure 5 membranes-12-00573-f005:**
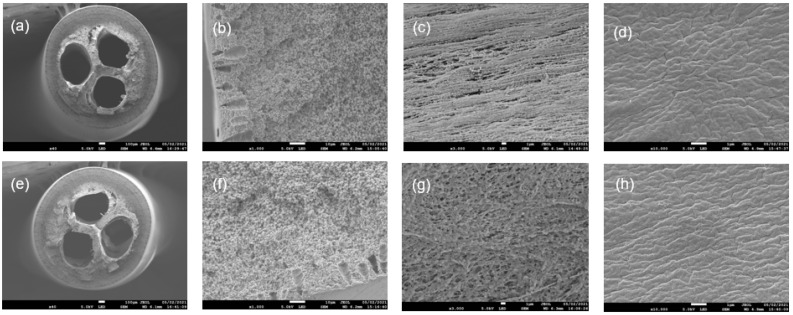
FESEM images of tri-bore PVDF hollow fiber membranes fabricated from batch sizes of (**a**–**d**) 20 kg and (**e**–**h**) 50 kg. (**a**,**e**) Cross section, (**b**,**f**) cross section at higher magnification, (**c**,**g**) inner surface, (**d**,**h**) outer surface. The PVDF surfaces were sputter-coated with platinum before measurement.

**Figure 6 membranes-12-00573-f006:**
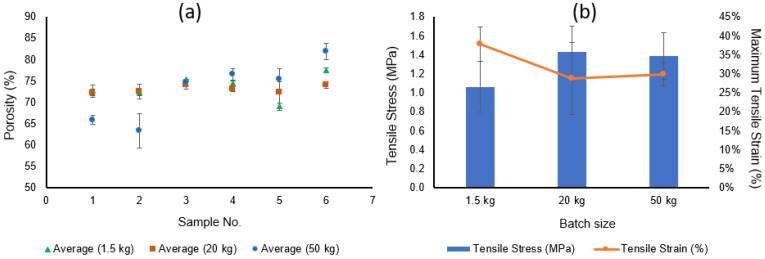
Comparison charts for (**a**) bulk porosity and (**b**) tensile properties of the PVDF TBHF membranes fabricated from 1.5 kg, 20 kg, and 50 kg dope batches.

**Figure 7 membranes-12-00573-f007:**
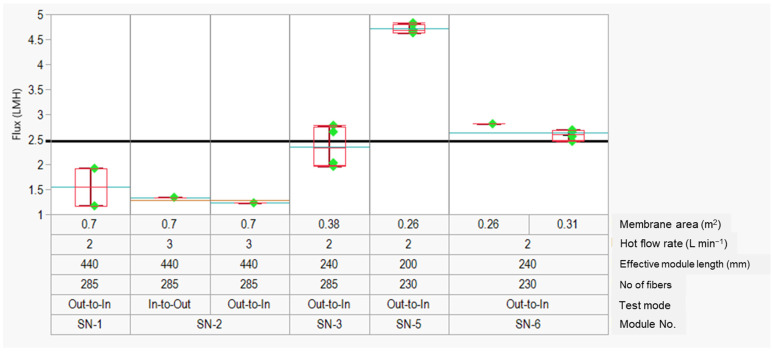
Flux optimization data for the PVDF TBHF membranes under DCMD mode with different module parameters and test conditions. The red boxes indicate the data range for the specific test conditions, whereas the blue lines indicate the group averages for the specific test conditions. The solid black line at 2.5 L m^−2^ h^−1^ indicates the population mean.

**Figure 8 membranes-12-00573-f008:**
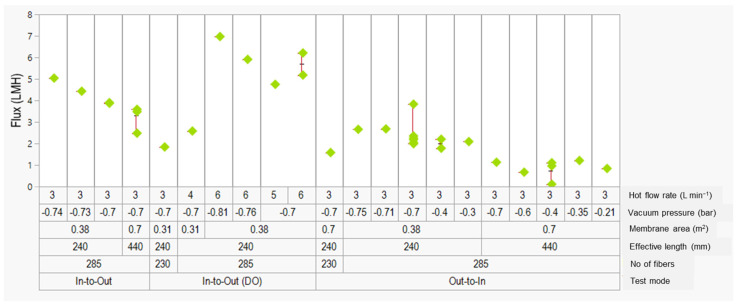
Flux optimization data for the PVDF TBHF membranes under VMD mode with different module parameters and test conditions. The flux data are categorized into group 1 (in-to-out flow), group 2 (in-to-out double-outlet flow), and group 3 (out-to-in flow).

**Table 1 membranes-12-00573-t001:** Dope composition and spinning conditions for the PVDF tri-bore hollow fiber membranes.

Dope Parameters	
Composition	wt.%
PVDF	14.50%
EG	10.50%
LiCl	5.0%
DMAC	70.0%
Tank temperature (°C)	80
Flow rate (Hz)	44.00
Mass flow rate (g min^−1^)	47.88
Tank pressure (bar)	7
Bore Liquid Parameters	
Composition: DMAc:DI water	72.5:27.5 (wt.%)
Flow rate (Hz)	4.8
Actual mass flow rate (g min^−1^)	4.6
Temperature (°C)	60
Process parameters	
Outlet coagulant	Water
Coagulation bath temp (°C)	Ambient (~25 °C)
Air gap (mm)	30
Take-up speed	Free-fall
Winder speed (rpm, set value)	7.8
Gelation (%)	90
Wheel (%)	86.5

**Table 2 membranes-12-00573-t002:** Testing conditions of DCMD and VMD for 0.5-inch and 2-inch modules.

Test Conditions/ Configuration	DCMD		VMD			
Test Mode	Out-to-In	Out-to-In	In-to-Out		Out-to-In	
Module diameter (inches)	0.5	2.0	0.5	2.0	0.5	2.0
Number of fibers	6	230–285	10	230–285	10	230–285
Effective length (mm)	100	200–440	120	200–440	120	200–440
Effective membrane area (m^2^)	0.002	0.260–0.7	0.007–0.009	0.38	0.007 0.009	0.38–0.7
Packing density (%)	20.0–24.0	35.2	24.0–34.0	35.2–43.5	24.0–34.0	35.2–43.5
Feed water (hot) flow rate (L min^−1^)	0.50–0.55	2	0.50–0.60	5.0–6.0	0.50–0.60	5.0–6.0
Cold water flow rate (mL min^−1^)	24	0.50–0.75	n/a	n/a	n/a	n/a
Feed (hot) water temperature (°C)	66–70	~80	80–84	84–90	80–84	84–90
Permeate (cold) water temperature (°C)	10–13	10–15	n/a	n/a	n/a	n/a
Vacuum pressure (bar)	n/a	n/a	−0.7 to −0.8	−0.7 to −0.8	−0.7 to −0.8	−0.7 to −0.8
Test duration (h)	1.5–2.0	1.5–2.0	1	>1	1	>1
Feed concentration (g L^−1^)	35	35	35	35	35	35

**Table 3 membranes-12-00573-t003:** Characterization and DCMD performance of membranes prepared from different batch sizes. DCMD flux and salt rejection were determined using 0.5-inch modules under out-to-in mode and a feed solution of 35 g L^−1^ NaCl.

	Batch Size (kg)		
Membrane Property (Units)	1.5 kg Batch	20 kg Batch	50 kg Batch
OD (mm)	1.81 ± 0.14	1.72 ± 0.18	1.92 ± 0.11
ID (mm)	0.41 ± 0.07	0.40 ± 0.06	0.48 ± 0.03
Contact angle (°)	87.5 ± 8.0	89.8 ± 4.5	83.2 ± 3.2
Liquid entry pressure (bar)	2.8 ± 0.2	4.2 ± 1.7	3.4 ± 0.4
Average pore diameter (μm)	0.434	0.257	0.315
DCMD flux (L m^−2^ h^−1^)	12.7 ± 1.2	11.9 ± 1.1	10.6 ± 0.6
Salt rejection (%)	≥99.9%	≥99.9%	≥99.9%

**Table 4 membranes-12-00573-t004:** DCMD and VMD performance of 0.5-inch and 2-inch modules consisting of PVDF TBHF membranes prepared from 50 kg batches. Tests were carried out in out-to-in mode using 35 g L^−1^ NaCl as the feed and DI water as the permeate-side liquid in DCMD.

	0.5-Inch	2-Inch
DCMD Flux (L m^−2^h^−1^)	10.6 ± 0.6	1.4 ± 0.4
Rejection (%)	≥99.9	≥99.9
VMD Flux (L m^−2^h^−1^)	12.4 ± 0.9	1.1 ± 0.1
Rejection (%)	≥99.9	≥99.9

**Table 5 membranes-12-00573-t005:** Specifications of module variables for flux optimization under DCMD and VMD testing modes using 2-inch modules.

Sample No.	Module Diameter (inches)	No. of Fibers	Effective Length (mm)	Surface Area (m^2^)
SN-1	2	285	440	0.7
SN-2	2	285	440	0.7
SN-3	2	285	240	0.38
SN-4	2	285	240	0.38
SN-5	2	230	200	0.26
SN-6	2	230	240	0.31
SN-7	2	220	240	0.30
SN-8	2	220	240	0.30

## Data Availability

Not applicable.

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
