# Peer review of "Scale Up and Validation of Novel Tri-Bore PVDF Hollow Fiber Membranes for Membrane Distillation Application in Desalination and Industrial Wastewater Recycling"

_membranes, 2022, doi:10.3390/membranes12060573_

Round 1

Reviewer 1 Report

Review of membranes-1739267

This manuscript is very well designed and executed. It also tackles the problem of scaling up from bench scale to industrial scale (20 and 50 kg of highly viscous polymeric dope solutions) of producing hollow fiber membranes for membrane distillation applications at various modes. Fabricating hollow fiber with one borehole is already challenging, but the authors able to produce industrial-scale hollow fiber membranes with three-bore holes with consistent morphology, structures, etc. There are however some items that have to be corrected, as follows:  

  1. Units must be written in consistent way. Since the major parameter discussed in this manuscript is the flux, with the unit of L m-2 h-1, then other units need to follow.
  2. Line 187: What does it mean with “PVDF 1”? Is it 1 kg, or 1 L, or other unit? Please kindly revise.
  3. Line 58-60: This is a paragraph with one sentence only. Please merge it with the previous or the next paragraph.

  1. Line 127: g L-1
  2. Line 135: dimethylacetamide --> with lowercase d --> this is suggested in order to be consistent with other chemicals that are begun with lowercase letters.
  3. Line 187: Separate “25” with “Hz”.
  4. Line 222: N m-1
  5. Line 229: mm min-1
  6. Line 241: g cm-3
  7. Table 2: L min-1; mL min-1; g L-1.
  8. Line 404-405: This is a paragraph with one sentence only. Please merge it with the previous or the next paragraph.
  9. Line 423: 1.5 kg --> with lowercase k
  10. Legend of Figure 6a: 1.5 kg; 20 kg; 50 kg --> with lowercase k
  11. Figure 6b, X-axis: 1.5 kg; 20 kg; 50 kg --> with lowercase k
  12. Line 489: g L-1
  13. Table 3: Please construct the table using the font type and font size required by the journal.
  14. Table 3, row 1 and row 2: kg --> with lowercase k
  15. Table 3, row 2, column 4: 50 kg --> not “50 g”
  16. Line 500: Delete the space between “90” and “ ° ” (degree sign)
  17. Line 521: g L-1
  18. Line 622: L min-1

  1. References: For the digital object identifier (DOI), please write the DOI of every paper in a uniform format. Currently, the DOI numbers are written not in a uniform manner (“DOI”, “Doi”, “doi”, “https://doi.org”, “doi.org/xxx”, “dx.doi.org/xxx”, “doi.org// --> even with double slashes”
  2. Reference 2: Delete “764” from the title.
  3. Reference 4: Delete “768” from the title. The publication year of this paper is 2021, not 2022.
  4. Reference 7: Double slashes only exist after “https:”
  5. Reference 19: Delete “809” from the title.
  6. Reference 21: Delete “779” from the title.
  7. Reference 22: Do not use “et al.”, please write the author list in complete manner. There are only five authors in this reference, please write them all.
  8. Reference 24: Delete “816” and “817” from the title.
  9. Reference 29: …van der Bruggen, B.--> NOT Bruggen, B.v.d
  10. Reference 43: …Zhang, S. --> uppercase S, not lowercase s.
  11. Reference 2: Delete “764” from the title.
  12. References from “Water Science and Technology” must be abbreviated as “Water Sci. Technol.”
  13. References from “Separation and Purification Technology” must be abbreviated as “Sep. Purif. Technol.”
  14. Caption of Figure S2: need a revision, please check.
  15. Caption of Table S1: 1.5 kg --> with lowercase k.
  16. Please upload again the revised supplementary materials separately.

Author Response

Thank you for the valuable feedback. Please see the attachment for our responses to your insightful suggestions. 

Reviewer 2 Report

The manuscript described how to scale up PVDF hollow fiber membrane for membrane distillation, with a great deal of parameters provided and analyzed. Instead of a research paper, it mostly likes a scientific report. However, considering a broad reader's interest, the reviewer would like to suggest considering this work to be published after addressing the following points.

1. Anti-fouling properties of the hollow fiber membrane should be characterized since it is very important for a real application.

2. A long-term distillation performance should be demonstrated, for example, one month or two, for the applications in seawater and wastewater treatments, to verify the possibility for a practical application.

3. The purification efficiency, either for seawater or wastewater treatment, should be compared with some commercially available counterparts, to exhibit the advantages of the PVDF hollow fibers over other membranes. 

Author Response

Thank you very much for your valuable suggestions. Please see the attached file for our responses and clarifications. 

Thank you 

Reviewer 3 Report

What was the shape of the HF take-up wheel?. And was the speed of rotation compensated during the winding of successive layers of HF?

  In the discussion (section 3.2.1) the differences in the obtained membranes resulting from the winding process on a 2 m diameter wheel are described. The shape is also not given here. And whether the “wheel” is round or hexagonal is essential!

 In the SEM description (Fig. 5) there is no data whether the photomicrographs were made with the use of sputtering with the conductor (what?) Or without sputtering with the conductor. The lack of this information makes it impossible to evaluate the presented photomicrographs.

Author Response

Thank you very much for the valuable comments and suggestions. Please see the attached file for our responses and clarifications. 

Thank you 
